# Imaging timing after glioblastoma surgery (INTERVAL-GB): protocol for a UK and Ireland, multicentre retrospective cohort study

Conor S Gillespie [1,2] Emily R Bligh,[3] Michael T C Poon,[4,5] Georgios Solomou [6] Abdurrahman I Islim,[7,8] Mohammad A Mustafa,[1,2] Ola Rominiyi,[9,10] Sophie T Williams,[9,11] Neeraj Kalra,[12] Ryan K Mathew [12,13] Thomas C Booth,[14,15] Gerard Thompson,[16,17] Paul M Brennan,[5,16] Michael D Jenkinson,[1,2] INTERVAL-GB Collaborative,[18] British Neurosurgical Trainee Research Collaborative (BNTRC),[18] Neurology and Neurosurgery Interest Group (NANSIG)[19]

CSG and ERB are joint first authors.

For numbered affiliations see end of article.

**Correspondence to**
Dr Conor S Gillespie;
hlcgill2@liv.ac.uk

## ABSTRACT

**Introduction** Glioblastoma is the most common malignant primary brain tumour with a median overall survival of 12–15 months (range 6–17 months), even with maximal treatment involving debulking neurosurgery and adjuvant concomitant chemoradiotherapy. The use of postoperative imaging to detect progression is of high importance to clinicians and patients, but currently, the optimal follow-up schedule is yet to be defined. It is also unclear how adhering to National Institute for Health and Care Excellence (NICE) guidelines—which are based on general consensus rather than evidence—affects patient outcomes such as progression-free and overall survival. The primary aim of this study is to assess MRI monitoring practice after surgery for glioblastoma, and to evaluate its association with patient outcomes.

**Methods and analysis** ImagiNg Timing aftER surgery for glioblastoma: an eVALuation of practice in Great Britain and Ireland is a retrospective multicentre study that will include 450 patients with an operated glioblastoma, treated with any adjuvant therapy regimen in the UK and Ireland. Adult patients ≥18 years diagnosed with glioblastoma and undergoing surgery between 1 August 2018 and 1 February 2019 will be included. Clinical and radiological scanning data will be collected until the date of death or date of last known follow-up. Anonymised data will be uploaded to an online Castor database. Adherence to NICE guidelines and the effect of being concordant with NICE guidelines will be identified using descriptive statistics and Kaplan-Meier survival analysis.

**Ethics and dissemination** Each participating centre is required to gain local institutional approval for data collection and sharing. Formal ethical approval is not required since this is a service evaluation. Results of the study will be reported through peer-reviewed presentations and articles, and will be disseminated to participating centres, patients and the public.

## STRENGTHS AND LIMITATIONS OF THIS STUDY

⇒ The study will include a large cohort of approximately 450 patients.
⇒ The short study eligibility period (6 months) will facilitate inclusion from over 20 neuro-oncology centres in the UK and Ireland.
⇒ Use of a mandatory training module will enable standardised data collection, with clarity of definitions.
⇒ The retrospective study design will increase case numbers and statistical power, but may lead to selection and information bias.
⇒ Study results may not be fully applicable to studies outside the UK and Ireland, or centres with limited access to MRI.

## INTRODUCTION

Glioblastoma is the most common malignant primary brain tumour,[1] with an annual incidence of 5 per 100 000 people.[2 3] In the UK, almost 3350 new diagnoses of glioblastoma are made every year.[4] Glioblastoma are aggressive and incurable. The median overall survival (OS) is 6–17 months,[5] extended for those treated with standard care,[6] a combination of maximal tumour debulking, radiotherapy and chemotherapy.[6–8] Glioblastoma is currently incurable due to their infiltrative nature (which means the disease burden can never be completely surgically cleared) and treatment resistance. Progression of residual is therefore inevitable.[9 10]

After primary treatment, patients are followed up with serial surveillance MRI to detect disease progression. The current National Institute for Health and Care Excellence (NICE) guidelines recommend

considering a postoperative scan to assess extent of resection within 72 hours of surgery, scans at every 3–6 months for the first 2 years after finishing treatment, followed by every 6–12 months for the first 5 years, then 1 to 2 yearly imaging for life. These NICE guidelines are based on general consensus, due to a lack of available evidence and differ from European Association of Neuro-Oncology guidelines, which advocate performing an MRI scan within 24–48 hours postoperatively.[11] In addition to routine scheduled imaging, MRI after surgery may also be indicated when clinical symptoms suggest disease progression (symptomatic/unscheduled).[12 13] A recent Cochrane review identified little evidence for an optimal imaging strategy.[12] Furthermore, a survey of neuro-oncology centres in the UK (GIN-CUP study) demonstrated substantial heterogeneity in imaging practices after surgery,[14] and timing of imaging has not been previously described.[15]

In the James Lind Alliance Priority Setting Partnerships, the effect of interval scanning on detection of tumour progression and survival was identified as a high priority for research.[16–18] This was due to the patient perspective of increased scanning frequency being associated with earlier progression detection, initiation of treatment and potential survival improvement. A recent position statement by the National Cancer Research Institute (NCRI) Brain Tumour group emphasised the importance of appropriately powered studies that examine imaging frequencies after surgery for glioblastoma, in order to address the lack of evidence evaluating the optimal scanning method, and if scheduled and triggered imaging should be employed in clinical practice.[19] There is little individual-level patient data on imaging practice and if we are to optimise the schedules for brain imaging, we need to first determine current practices in the UK and Ireland. Our study addresses this.

## Objectives

The primary objective of this study is to describe MRI surveillance practice after surgery for patients with glioblastoma who have received adjuvant oncology treatment. The secondary study objectives are to assess indications for scanning patients during the postoperative follow-up period, and to compare progression-free survival (PFS) and OS in patients who were and who were not scanned in accordance to NICE guidelines.

## METHODS AND ANALYSIS
### Study design

This is a national (UK and Ireland) multicentre retrospective study, for all UK and Ireland neuro-oncology units and associated neurosurgery services. Data will be collected on consecutive surgical patients with a new histopathological diagnosis of glioblastoma, with surgery between the dates of 1 August 2018 and 1 February 2019 (6 months) (tables 1 and 2). Local investigators will identify eligible patients using existing medical and radiological scan records. The study will assess routine clinical practice without change to patient care, so this study requires local institutional approval in each participating unit for data collection and sharing, but not National Health Service (NHS) research ethics committee (REC) approval. The study has been approved by the Walton Centre NHS Foundation Trust hospital audit committee (approval number NS 370). The study is led by the Neurology and Neurosurgery Interest Group (NANSIG), an international student and junior doctor led collaborative interest group (www.nansig.org),[20 21] and supported by the British Neurosurgical Trainee Research Collaborative.[22] The study data collection period will commence on Monday 15 November 2021, and will end on Thursday 21 July 2022.

**Table 1** Summary of recorded study variables

| Recorded variable | Description |
| --- | --- |
| Baseline clinical and radiological | Age; date of surgery; sex; WHO performance status; presence of preoperative seizure activity; preoperative neurological deficit. |
| Radiological | Location; laterality; main anatomical area. |
| Surgical and histopathological | Extent of resection (gross total resection, subtotal resection or biopsy). |
| Histopathology | IDH status; MGMT promoter status. |
| Adjuvant treatment | Adjuvant radiotherapy start date; end date; radiotherapy dose and fractions; number of adjuvant temozolomide cycles completed. |
| Other supportive treatment | Enrolment into a clinical trial; reoperation for tumour; second-line or third-line chemotherapy; fourth-line chemotherapy; reirradiation; Referral to palliative care. |
| Follow-up imaging | Date of first postoperative MRI and indication; date of subsequent scans; if scans were unscheduled or scheduled; scan outcomes. |
| Outcome measures | Disease progression; overall survival. |

All definitions can be found in the main text, and online supplemental appendix A
IDH, isocitrate dehydrogenase; MGMT, O6-methylguanine-DNA methyltransferase.

**Table 2** Study (Gannt) flowchart

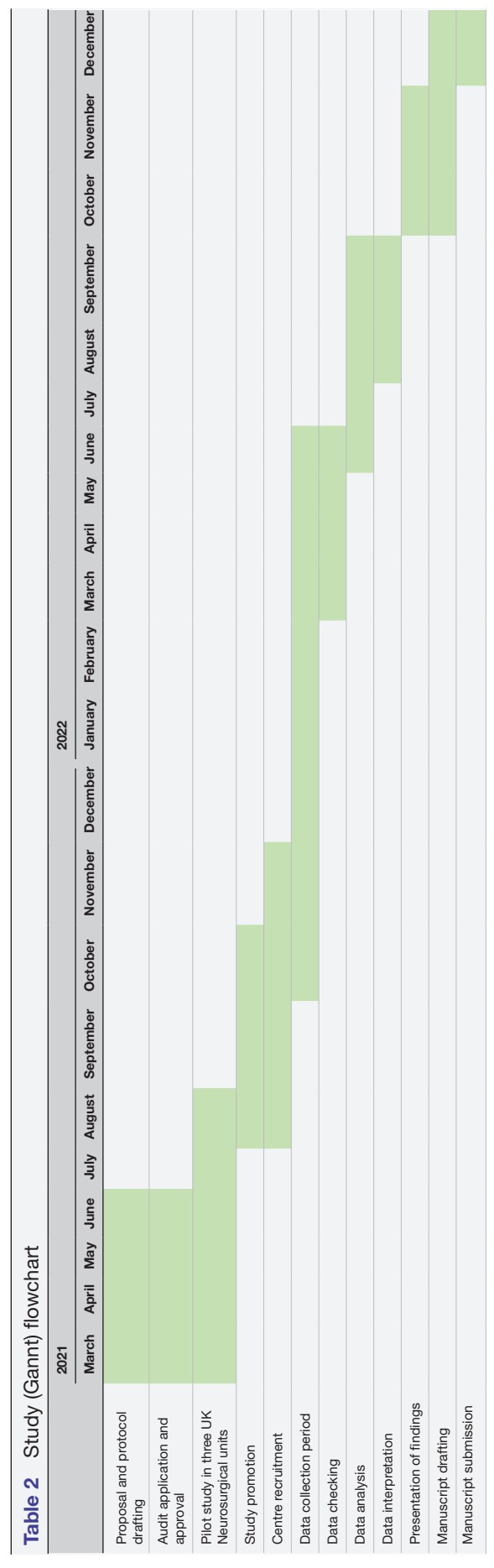

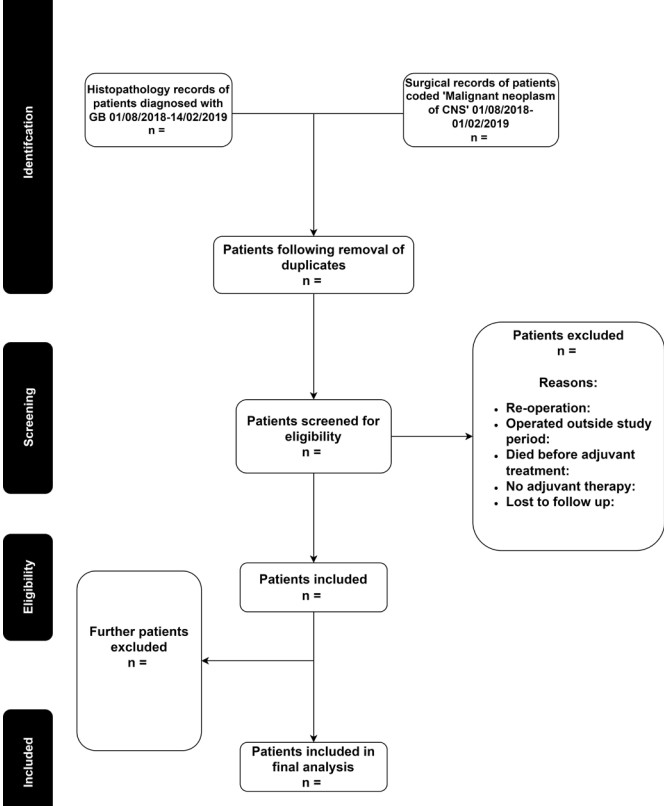

**Figure 1** Process of identifying a patient list at each collaborating neurosurgical unit. CNS, Central Nervous System; GB, Glioblastoma.

### Study population and eligibility criteria

The study will include adults (≥18 years) newly diagnosed with glioblastoma, as defined by a consultant histopathologist, undergoing surgery (including biopsy) between 1 August 2018 and 1 February 2019 with any concomitant and/or adjuvant therapy, or radiotherapy alone (including reduced dose and palliative intention). Patients must have at least one postoperative MRI scan in relation to their glioblastoma diagnosis to be eligible. Patients will be excluded if they have no surgical intervention, if they had no adjuvant treatment, had no established histopathological diagnosis, follow-up with CT only, or unavailability of medical notes to ascertain follow-up. Patients who did not receive adjuvant treatment will be excluded- this is to exclude patients that solely underwent a diagnostic biopsy, without intent of increasing OS or PFS.

### Patient identification

Local investigators will use the surgical and neuropathology records, hospital discharge codes and multidisciplinary team (MDT) documents to identify potentially eligible patients (figure 1). Investigators will assess each patient's eligibility against the inclusion and exclusion criteria.

### Sample size

The sample size calculation was derived from the results of a multicentre pilot study, including three participating

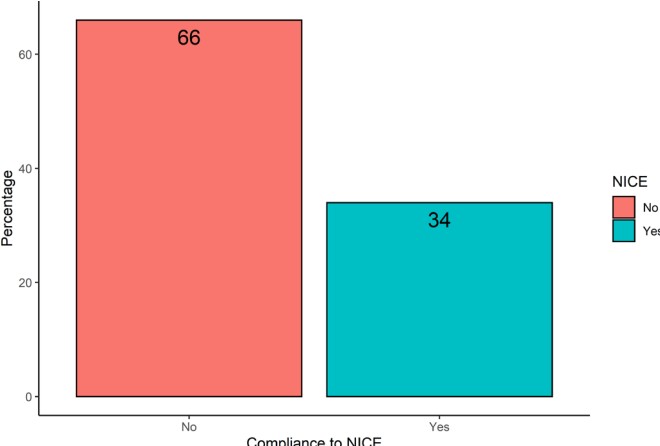

**Figure 2** Bar chart representing pilot data compliance to NICE guidelines. NICE, National Institute for Health and Care Excellence.

centres. The pilot included 123 patients, with a median survival of 9.6 months. The proportion of patients who had imaging scheduled in accordance with NICE recommendations was 34% (figure 2). The null hypothesis of the study is that more than half of patients (>50%) will not be scanned in accordance with NICE guidelines, and that groups scanned according to these guidelines will equate to improved OS (superiority assumption), and shorter PFS (due to recurrence being detected earlier due to more intense imaging surveillance). Therefore, we chose a HR of 1.35 as our minimally clinically important difference for the study, which roughly equates to 3 months of survival benefit. The margin is similar to the survival benefit of patients undergoing temozolomide therapy.[6] Assuming a mortality rate of 85% of patients during the 24-month follow-up period, to achieve 80% power, with a 5% type 1 error rate, the sample size required for the ImagiNg Timing aftER surgery for glioblastoma: an eVALuation of practice in Great Britain and Ireland (INTERVAL-GB) study is 456 patients (approximately 22 Neurosurgical units, assuming an average patient list of 20 patients per neurosurgical centre).

## Data collection
Data will be collected at each centre by members of the local collaborating team. Data will be collected from a combination of the patient's clinical, radiological and histopathological records (coding for 'glioblastoma' or ICD-10 code 'C71.9- Malignant neoplasm of brain'), to obtain data points relating to surgery, adjuvant treatments and MRI follow-up. Data collection fields are outlined in online supplemental appendix A, and these relate to routinely available patient, tumour, molecular details, follow-up imaging and survival data until date of death, or last follow-up (last day of data collection period).

Each neuro-oncology centre will have a local collaborative team formed by the INTERVAL-GB local lead, local collaborators, a neurosurgical trainee and a consultant neurosurgeon. The INTERVAL-GB local teams must be supported by a trainee and Consultant Neurosurgeon, or Clinical Oncologist to ensure quality of data collected is maintained. The INTERVAL-GB local lead will be responsible alongside local collaborators for data entry on the online database. Local teams are advised that if radiological data, such as tumour location, is not clear from the radiology report, clarification should be sought from the neurosurgical trainee, and reviewed by the consultant neurosurgeon, radiologist or oncologist if there is still any doubt. If there is any uncertainty with data, the local team must also clarify with the trainee, consultant, radiologist or oncologist. Data collaborators will be asked not to interpret scans and are instead advised to refer to the radiologist's report, and MDT reports to define progression.

## Castor database
Data will be collected locally and submitted to an online server on the secure database—Castor (Castor, New York, USA). Castor is a secure web application used to manage online study databases. Co-leads for the study (CSG and ERB) have overseen the development of a data collection tool (online supplemental appendix A) which is accessible via any electronic device with internet access. Castor databases comply with the UK Data Protection Act 2018 and the European Union General Data Protection Regulation (GDPR). Each patient will have a unique study number traceable to the identifiable patient information only through forms stored physically on NHS sites, or on password-protected NHS computers or servers.

## Recorded variables
### Baseline clinical and radiological variables
Recorded variables, and their definitions, are listed in table 1 and online supplemental appendix A. These include Age at date of surgery, sex, WHO performance status, presence of pre-operative seizure activity (defined as any seizure occurring within 12 months of diagnosis and before surgery), presence of preoperative neurological deficit (defined as any deficit affecting sensory, motor, or cranial nerve function). Baseline radiological variables will include location of GBM, laterality and main anatomical area involved (online supplemental appendix A).

### Surgical and histopathological variables
Extent of resection (gross total resection (GTR), subtotal resection (STR) or biopsy) will be categorised according to the following criteria: GTR is defined as no residual contrast-enhancing tumour on T1-weighted, contrast enhanced MRI within 48–72 hours postoperatively, as judged by the neuroradiologist and the neurosurgeon (ie, complete resection). If any residual contrast-enhancing tumour is present, it is classified as a STR. If a tumour has been less than 50% removed, or labelled as a biopsy, it will be considered a biopsy. Isocitrate dehydrogenase status (wild-type, mutant or not recorded) and O6-methylguanine-DNA methyltransferase promoter

status (unmethylated, methylated, not done, test failed/insufficient sample quality) will be recorded.

### Adjuvant and supportive treatment details

Radiotherapy (start date, end date, total radiotherapy dose (Gy), fractions) and chemotherapy details (drug and number of cycles), enrolment into a clinical trial, name of trial if applicable, reoperation for tumour (with date of surgery, and extent of resection recorded), second, third or fourth line chemotherapy (with details of drug, start date, end date, dose, number of cycles) and reirradiation if applicable (start date, end date, total radiotherapy dose (Gy), fractions) will be recorded. Referrals to palliative care for patients who have had any adjuvant oncology treatment will be noted.

### Follow-up imaging details

Date of first postoperative MRI and the indication (neurosurgical to assess extent of resection or radiotherapy planning), if MRI was completed within (1) 72 hours of surgery (yes/no) and (2) 48 hours of surgery, date of subsequent MRI scans and whether they were scheduled or unscheduled will be recorded. Unscheduled will be defined as a scan performed due to clinical symptoms, deterioration or emergency presentation. A scan outcome will be classified as stable disease, progressive disease, or pseudo-progression. If progressive disease, the MDT outcome associated with the scan will be reviewed and categorised into progressive disease confirmed or not confirmed.

### Outcome measures

The study outcome measures will be OS, defined as the time in months from the date of surgery to the date of death as an endpoint and disease progression, defined as time in months from the date of surgery to the date of scan demonstrating progressive disease confirmed as per MDT.

### Data validation and quality assurance

Primary investigators and the data collection team will meet midway through the data collection phase, and at completion of data collection, to identify any issues or concerns. The trainee collaborator is responsible for providing clarification for data entry points, or answering queries that the INTERVAL-GB local lead and local collaborators may have with the data collection process. It is the responsibility of the INTERVAL-GB local lead to arrange these meetings. Before commencement of data collection, all student and junior doctor collaborators (local lead, and all data collaborators who are not currently in neurosurgery or oncology specialist training) will need to attend a 60 min, online mandatory training module, hosted virtually over Zoom. This encompasses an introduction to glioblastoma and glioblastoma management, the evidence gap, the study protocol, a comprehensive live demonstration of the Castor data collection tool, and time to allow students and junior doctors to ask questions before commencement of the study itself. Each collaborator will be required to complete an online form, only available after attending a mandatory training session, to evidence that as a data collector they have completed mandatory training.

Each trainee will be involved in the validation of local data collected by the NANSIG collaborator. After data have been completed for a participating centre, any scan results, or MDT results that are categorised as 'unclear' by the NANSIG collaborator, will be reviewed, validated, and reasons why provided by the trainee, to ensure internal validation of data ascertained. Trainee collaborators will also support day-to-day data collection by the local student lead should queries arise.

Five neuro-oncology centres will also carry out data validation. This will consist of an external team of data validators (same neurosurgical unit, who have not been involved in data collection or entering patient data), collecting and uploading the same patient information to Castor. This duplicate data will be removed after the study end date, and each data point analysed for similarity. We will accept a 95% similarity or greater between the data collection teams for each data point. If significant discrepancies are detected beyond this, the neurosurgical trainee or consultant will be asked to review each case to reach a consensus.

### Statistical analysis

#### Descriptive analysis

Description of study data will be proportions for categorical variables and median with interquartile ranges for continuous variables. The prespecific subgroup included: patients whose imaging schedule was entirely concordant with NICE recommendations and patients whose imaging schedule had any deviation from NICE recommendations. The definition of concordance to NICE recommendations is defined as having follow-up scans every 3–6 months after finishing treatment (for the first 2 years of follow-up). Descriptive statistics will be generated to describe the primary objective of the study, according to this definition. Descriptive statistics will also be presented by this subgroup (a group that followed the guidelines and the group that did not have follow-up scans every 3–6 months). For the description of overall and PFS, we will use the Kaplan-Meier estimator to generate the medians by NICE concordant subgroups using the right-censored survival data, and use this method to compare subgroups with the log-rank test, followed by multivariable Cox-regression survival analysis to confirm statistically significant ($p<0.05$) findings. Patients without complete records will be excluded from statistical analysis.

### Patient and public involvement

Public and patient involvement was attained for 'INTERVAL-GB' and played a role in the design and conception of the study. The 'INTERVAL-GB' study leads contacted The Brain Tumour Charity, and two representatives voluntarily agreed to provide insight and feedback on our proposed study from the viewpoint of somebody with a spouse who died as a result of a glioblastoma. We held

a 1-hour open video discussion with one representative, while the other provided us detailed written feedback via email. Following discussion and reflection, it was decided to refine the data collection proforma, to collect more data specifically targeted at palliative care enrolment including chemotherapy end dates, date of palliative care enrolment and at a small number of centres—the type of palliative care received, for example, hospice or community. The aim of these amendments was to increase study focus on patient quality of life. It was felt by both representatives that waiting for MRI results was an anxiety building process for the patient and their family, as such finding time from scan to communicated result would be valuable to the glioblastoma community, however, collecting this data was not deemed feasible retrospectively and would be more accurately ascertained in a prospective study.

## ETHICS AND DISSEMINATION
### Study registration
The local lead and accompanying research team at each unit are responsible for registering the study as a retrospective observational study with the clinical audit department of their respective centre, including Caldicott guardian and information governance approval as required. Local leads should send proof of local audit and data approvals to the primary investigators on registration before starting data collection.

### Local investigator responsibilities
Each local NANSIG student investigator will be responsible for the overall study conduct and compliance with the protocol. The investigator must have read and familiarised themselves with the protocol and the study requirements, as evidenced by attendance to the training programme. All assisting staff (such as supervising consultants and trainees) should be informed of the protocol and its availability for review. The local NANSIG student lead at each centre is responsible for the quality of data recorded in the database. Each local NANSIG investigator will be asked to identify what trust systems they will need to access to complete data collection (including access to imaging reports, neurosurgical and oncology records) before commencement of the project, at their local centre).

### Confidentiality and data protection
No patient identifiable information will be uploaded or stored on the Castor database. The clinical investigators can only view the records from their own centres. All patient records must be kept in a manner designed to protect patient confidentiality in secure storage with limited access, such as in a password-encrypted database if stored online, or in a locked secure file in the case of physical notes. All data obtained should only be disclosed and used for the purposes of this study. This must also comply with the requirements of the Data Protection Act 2018 and GDPR according to latest legislation. Training on this is compulsory for local NANSIG student investigators and will be provided with access to an MRC Online module. Access to overall records from every site will be restricted to the study leads (ERB and CSG). Local data will be held for 2 years after the end of the study period, for clarification of data entries if required. Combined, anonymised data will be held on an encrypted, password protected, NHS Trust computer belonging to the study leads (ERB and CSG) for 15 years after study completion, to allow for further analysis and additional collaborations. After 15 years, the data file will be automatically deleted. Individual patient data will be anonymised before being accessed by the study authors, as part of institutional approval.

### Ownership
Control of the complete dataset arising from this study resides with the steering committee (named in the protocol). Control of local data rests with the local audit team. Proposals to use the data are welcomed and should be directed to the primary investigators, who will review the request alongside the steering committee.

### Dissemination of results
The results of the study will be presented by the steering committee at national and international scientific meetings, and will be submitted for publication in peer-reviewed journals. The study results will be reported using the Strengthening the Reporting of Observational studies in Epidemiology (STROBE) guidelines.

**Author affiliations**
[1]Institute of Systems, Molecular and Integrative Biology, University of Liverpool, Liverpool, UK
[2]The Walton Centre NHS Foundation Trust, Liverpool, UK
[3]Department of Neurosurgery, Institute of Neurological Sciences, Glasgow, UK
[4]Usher Institute, The University of Edinburgh, Edinburgh, UK
[5]Translational Neurosurgery, Centre for Clinical Brain Sciences, University of Edinburgh, Edinburgh, UK
[6]School of Clinical Medicine, University of Cambridge, Cambridge, UK
[7]Department of Neurosurgery, Manchester Centre for Clinical Neurosciences, Manchester, UK
[8]Division of Neuroscience and Experimental Psychology, Faculty of Biology, Medicine and Health, University of Manchester, Manchester, UK
[9]Sheffield Teaching Hospitals NHS Foundation Trust, Sheffield, UK
[10]Department of Neuroscience, The University of Sheffield, Sheffield, UK
[11]Department of Oncology and Metabolism, The University of Sheffield, Sheffield, UK
[12]Department of Neurosurgery, Centre for Neurosciences, Leeds General Infirmary, Leeds Teaching Hospitals NHS Trust, Leeds, UK
[13]School of Medicine, University of Leeds, Leeds, UK
[14]Department of Neuroradiology, King's College Hospital, London, UK
[15]School of Biomedical Engineering & Imaging Sciences, King's College London, London, UK
[16]Department of Clinical Neurosciences, NHS Lothian, Edinburgh, UK
[17]Edinburgh Neuro-oncology Translational Imaging Research (ENTIRe), Centre for Clinical Brain Sciences, University of Edinburgh, Edinburgh, UK
[18]Society of British Neurological Surgeons, London, UK
[19]Neurology and Neurosurgery Interest Group, London, UK

**Acknowledgements** We would like to thank all investigators, patients, and collaborators in helping to design, conceptualise, and contribute to the study.

**Collaborators** INTERVAL-GB Collaborative: Hidayatul Abdullmalek, Suhaib Abualsaud, Gideon Adegboyega, Chinelo Afulukwe, Ahmed Ahmed, Najma Ahmed, Michael Amoo, Abdelsalam Nedal Al-Sousi, Yahia Al-Tamimi, Ajitesh Anand, Prithvi Bahu, Neil Barua, Harsh Bhatt, Emily R Bligh, Peter Bodkin, Ion Boiangiu, Abbey Boyle, Christiaan Bredell, Paul M Brennan, Talhah Chaudri, Jeremy Cheong, Ana Cios, Iona Cleer, David Coope, Ian Coulter, Giles Critchley, Harriet Davis, Paolo Jose De Luna, Nayan Dey, Bea Duric, Abdullah Egiz, Justyna O. Ekert, Chinedu Brian Egu, Jinendra Ekanayake, Anna Elso, Tomas Ferreira, Tom Flannery, Kwan Wai Fung, Rahul Ganguly, Conor S Gillespie, Sanay Goyal, Emily Hardman, Lauren Harris, Theodore Hirst, Kelvin Sunn Hoah, Sam Hodgson, Kismet Hossain-Ibrahim, Lena Mary Houlihan, Sami Squali Houssaini, Sadid Hoque, Dana Hutton, Kismet Ibrahim, Abed Islim, Mahnoor Javed, Michael D Jenkinson, Neeraj Kalra, Siddarth Kannan, Efthymia Kapasouri, Andrew Keenlyside, Kristy Kehoe, Bharti Kewlani, Prerna Khanna, Rosaline de Koning, Kunalika Sathish Kumar, Ashvin Kuri, Simon Lammy, Eunkyung Lee, Robert Magouirk, Andrew Martin, Riccardo Masina, Ryan K Mathew, Adele Mazzoleni, Patrick McAleavey, Gráinne McKenna, Daniel McSweeney, Saad Moughal, Mohammad Arish Mustafa, Engelbert Mthunzi, Trinh Ton Nu Ngoc, Shiva A Nischal, Michael O'Sullivan, Jay J Park, Jonathan Pesic-Smith, Peter Peterson, Isaac Phang, Puneet Plaha, Michael T C Poon, Shyam Pujara, George E Richardson, Marwa Saad, Shinjan Sangal, Veekshith Shetty, Natalie Simon, Robert Spencer, Rosa Sun, Irtiza Syed, Jesvin Tom Sunny, Anca-Mihaela Vasilica, Armin Nazari, Daniel O'Flaherty, Setthasorn Zhi Yang Ooi, Arslan Raja, Daniele Ramsay, Renitha Reddi, Daniel Richardson, Elena Roman, Ola Rominiyi, Dorina Roy, Omar Salim, Jeremiah Samkutty, Jashan Selvakumar, Thomas Santarius, Avani Shanbhag, Stuart Smith, Agbolahan Sofela, Edward Jerome St. George, Preethi Subramanian, Vaibhav Sundaresan, Kieron Sweeney, Ronald Tan, Nicole Turnbull, Yuewei Tao, Lewis Thorne, Rebecca Tweedie, Anastasia Tzatzidou, Babar Vaqas, Sara Venturini, Kathrin Whitehouse, Peter Whitfield, Jack Wildman, Isabelle Williams, Karl Williams, Victoria Wykes, Tiffany Tze Shan Ye, Kelvin Sunn Yap, Mahir Yousuff, Asaad Zulfiqar. British Neurosurgical Trainee Research Collaborative (BNTRC) committee: Michael T C Poon, Abdurrahman I Islim, Angelos Kolias, Julie Woodfield, Aswin Chari, Neeraj Kalra, Soham Bandyopadhyay, Robin Borchert, Rory Piper, Daniel M. Fountain. Neurology and Neurosurgery Interest Group (NANSIG) core committee members: Conor S Gillespie, Emily R Bligh, Soham Bandyopadhyay, Jay J Park, Setthasorn ZY Ooi, George E Richardson, Abigail Clynch, Oliver Burton, Avani Shanbhag, Moritz Steinruecke, William Bolton, Bharti Kewlani, Alvaro Yanez Touzet, Hannah Redpath, Seong Hoon Lee, Abdullah Egiz, Joshua Erhabor, Orla Mantle. INTERVAL-GB Steering committee: Conor S Gillespie, Emily R Bligh, Michael T C Poon, Georgios Solomou, Melissa Gough, Abdurrahman I Islim, Christopher P Millward, Ola Rominiyi, Rasheed Zakaria, Paul M. Brennan, Stephen J. Price, Colin Watts, Sophie Camp, Thomas C Booth, Gerard Thompson, Samantha J Mills, Adam Waldman, Michael D Jenkinson.

**Contributors** CSG, ERB, GS, AII, MTCP and MDJ conceived the study. CSG and ERB drafted the initial study protocol. GS, AII, MAM, STW, OR, MTCP, NK, RKM, TCB, GT, PMB and MDJ provided input and suggestions for the final protocol. All authors proofread and approved the final manuscript.

**Funding** The authors declare a research bursary from the North West Cancer Fund, used to pay for the Data collection tool, Castor, for 6 months use (award number N/A). TCB declares funding from a Wellcome Trust/ESPRC grant (award number WT203148/Z/16/Z).

**Competing interests** None declared.

**Patient and public involvement** Patients and/or the public were involved in the design, or conduct, or reporting, or dissemination plans of this research. Refer to the Methods section for further details.

**Patient consent for publication** Not applicable.

**Provenance and peer review** Not commissioned; externally peer reviewed.

**ORCID iDs**
Conor S Gillespie http://orcid.org/0000-0002-9153-3077
Georgios Solomou http://orcid.org/0000-0002-9795-0517
Ryan K Mathew http://orcid.org/0000-0002-2609-9876

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
