## [Reviewer comments · BMJ Open]

ARTICLE DETAILS

TITLE (PROVISIONAL)	Imaging timing after glioblastoma surgery (INTERVAL-GB): protocol for a UK and Ireland, multi-centre retrospective cohort study
AUTHORS	Gillespie, Conor; Bligh, Emily; Poon, Michael; Solomou, Georgios; Islam, Abdurrahman; Mustafa, Mohammad; Rominiyi, Ola; Williams, Sophie; Kalra, Neeraj; Mathew, Ryan; Booth, Tom; Thompson, Gerard; Brennan, Paul; Jenkinson, Michael; Collaborative, INTERVAL-GB; Research Collaborative (BNTRC), British Neurosurgical Trainee; Interest Group, Neurology and Neurosurgery

VERSION 1 – REVIEW

REVIEWER	Yinzhong, Wang Lanzhou University First Affiliated Hospital, Department of radiology
REVIEW RETURNED	24-Apr-2022

GENERAL COMMENTS	To authors: The manuscript entitled "Imaging timing after glioblastoma surgery (INTERVAL-GB): protocol for a UK and Ireland, multi-centre retrospective cohort study" is very interesting. I would like to see the findings of the study that may benefit patients.
--

REVIEWER	Wang, Qun Chinese Peoples Liberat Army Gen Hosp, Neurosurgery
REVIEW RETURNED	26-Apr-2022

GENERAL COMMENTS	The authors performed a interesting MRI protocol (Imaging timing after glioblastoma surgery) to objectively assess MRI monitoring practice after surgery for glioblastoma, and to evaluate its association with patient outcomes. The manuscript is well written. And the study involved about 20 centers in UK and Ireland area. This manuscript could be considered for publishing after modification, before that, it still has a way to go. The weaknesses below need to be addressed to improve the quality of this manuscript. 1. The patients' number (approximately 450 patients) met the malignant brain tumor annual incidence 5/100,000, according the total UK and Ireland population (66,570,000,2018), it makes sense. However, the GBM incidence is lower than the reported
--

	data. It will be better to explain the data of different glioma type for the design section. 2. As a protocol study, the study design and methodology is very important, the study registration number should be mentioned, the number of NIH (http://clinicaltrials.gov/) or EU Clinical Trials Register (EU-CTR). 3. For the appendices, MRI device information and the including sequences (T2w, T1w, or DWI) should be clarified as a MRI imaging study.
--	--

REVIEWER	Korshoej, Anders Aarhus University Hospital, Department of Neurosurgery
REVIEW RETURNED	02-May-2022

GENERAL COMMENTS	The authors present a study protocol for a retrospective study across 20 UK centres in the period aug 2018 til feb 2019 with the aim of describing the MRI surveillance practice after surgery for patients with glioblastoma who have received adjuvant oncology treatment. The secondary study objectives include assesment of the indications for scanning patients during the post-operative follow up period, and comparison of progression-free survival and overall survival in patients who were and who were not scanned in accordance to NICE guidelines. Although a relevant descriptive study, its retrospective nature makes it less relevant to publish the protocol prior to data analysis. Comments:  1) The authors refer to the NICE guidelines stating that a post-operative MRI should be done within 72 hrs. Recent discussions have questioned this recommendation, see EANO guidelines 2021, suggested to conduct post-op scan within 48 hrs. The authors should mention and consider this in their study. 2) The authors chose to exclude all patients who did not receive adjuvant treatment. According to previous studies this is a significant proportions of those operated (around 25%). The authors should argue why these are excluded. I would assume all patients operated are relevant for the endpoints relating to post-op MRI, 3) Inclusion: The authors should state which codes that will be used to identify patients. 4) The authors do not define specific endpoint or the relating statistics. This makes it difficult to understand the statistical strategy that they will employ to achieve their primary and secondary aims. These aims are not related to specific statistics. The authors should design specific endpoints and statistica to measure and report their primary aims and hypotheses. 5) The authors do not present well-defined hypotheses or relating statistical hypotheses, except for a single metric on OS in the NICE group vs all others, which does not clearly address the stated aims. 6) It is unclear, whether the authors ahve designed their study as a superiority or non-inferiority study. Based on the current statistical description, it seems the authors designed their study to document study failure to 5% significance assuming a survival HR of 1.35 in the two groups (NICE vs non-NICE). This is equivalent to a failed superiority assumption of adherence to the NICE guidelines. Rather the authors claim that these guidelines would arguably not result in significant changes in HR and thus the study should be conducted as an equivalence or non-inferiority study. The authors
--

	should clarify this and also elaborate on the statistical section and design aspect considerably. 7) Regarding categorization of extent of resection, it is recommended that the authors adhere to RANO guidelines on post-op EOR.
--	--

VERSION 1 – AUTHOR RESPONSE

- Reviewer: 1

Comments to the Author:

- The manuscript entitled "Imaging timing after glioblastoma surgery (INTERVAL-GB): protocol for a UK and Ireland, multi-centre retrospective cohort study" is very interesting. I would like to see the findings of the study that may benefit patients.

Thank you very much for this comment. We hope that this study explores an important, yet under-researched area in glioblastoma, and hope the findings will help clarify imaging practice, for the ultimate benefit of patients.

Reviewer: 2

- The authors performed a interesting MRI protocol (Imaging timing after glioblastoma surgery) to objectively assess MRI monitoring practice after surgery for glioblastoma, and to evaluate its association with patient outcomes. The manuscript is well written. And the study involved about 20 centers in UK and Ireland area.

Thank you for this comment. We hope the large study size will provide evidence of national practice, and help to define future studies assessing optimal imaging paradigms.

- The patients' number (approximately 450 patients) met the malignant brain tumor annual incidence 5/100,000, according the total UK and Ireland population (66,570,000,2018), it makes sense. However, the GBM incidence is lower than the reported data. It will be better to explain the data of different glioma type for the design section.

Thank you for raising this important concern. The incidence is based on the referenced studies (Poon 2020 and Philips 2015) which both have glioblastoma incidence data directly available. For Philips, the 2015 GBM incidence in the UK was exactly 5/100,000, and, although Poon only reported an incidence of all glioma types to be 8.5/100,000, but of these, glioblastoma (grade IV glioma only) accounted for 47.9-52.1% of all glioma types included. This would give it an approximate incidence of 4.3/100,000, which would lead to similar numbers. We have modified the section to reflect the lower numbers.

- As a protocol study, the study design and methodology is very important, the study registration number should be mentioned, the number of NIH (<http://clinicaltrials.gov/>) or EU Clinical Trials Register (EU-CTR).

We thank the reviewer for this comment- as this study is a retrospective, observational study, and not a clinical trial, we have not registered the study on NIH or Clinical Trials Register. We hope that publication of the protocol in BMJ Open, will allow us to both publish a public record of the study, and ensure clarity with regards to adhering to the protocol and statistical analysis plan outlined.

- For the appendices, MRI device information and the including sequences (T2w, T1w, or DWI) should be clarified as a MRI imaging study.

Thank you for this comment- we have amended the appendices, to include imaging sequences in the data collection proforma. (Item 29)

Reviewer: 3

- Although a relevant descriptive study, its retrospective nature makes it less relevant to publish the protocol prior to data analysis.

Thank you- we firmly believe that the novelty, and topical nature of the research question merit publication of this protocol, to aid in disseminating the work and subsequent findings when complete.

- The authors refer to the NICE guidelines stating that a post-operative MRI should be done within 72 hrs. Recent discussions have questioned this recommendation, see EANO guidelines 2021, suggested to conduct post-op scan within 48 hrs. The authors should mention and consider this in their study.

Thanks for this comment- as the study is based in the UK and Republic Of Ireland, we adopted NICE guidelines as the benchmark for this study. We do recognise that the latest EANO guidelines differ, and therefore we have included this separate, within 48-hour definition, in our analysis plan. Additionally, we have updated our introduction to offer clarity on the differing guidelines.

- The authors chose to exclude all patients who did not receive adjuvant treatment. According to previous studies this is a significant proportions of those operated (around 25%). The authors should argue why these are excluded. I would assume all patients operated are relevant for the endpoints relating to post-op MRI,

Thank you for raising this important issue. We did indeed exclude patients who did not receive adjuvant treatment. This patient cohort were excluded because this group could include patients who received a biopsy, without any kind of treatment. With a biopsy being done for diagnosis and not to extend OS or PFS, there is less requirement to follow standardised imaging protocols. It was felt by the steering committee, which included Neurosurgeons, Clinical Oncologists, and Neuroradiologists, that excluding the biopsy population would be preferred, to avoid spurious imaging data (as these patients are less likely to be followed up according to guidelines). In addition, the NICE guidelines state that patients should be scanned '3-6 months after starting adjuvant treatment' therefore if patients did not receive this, they would not be eligible to comply with NICE guidelines. This has now been reflected in the study population and eligibility criteria paragraph.

- Inclusion: The authors should state which codes that will be used to identify patients. Thank you. We have now added the ICD-10 code that collaborators can use to identify patients, in Methods, data collection.

- The authors do not define specific endpoint or the relating statistics. This makes it difficult to understand the statistical strategy that they will employ to achieve their primary and secondary aims. These aims are not related to specific statistics. The authors should design specific endpoints and statistica to measure and report their primary aims and hypotheses.

- Thank you. We have now added date of death as a study endpoint, and included in descriptive analysis, is a point-by point-inclusion of statistics. We have also added that following a log-rank test, any differences should be confirmed with multivariable cox regression analysis.
- The authors do not present well-defined hypotheses or relating statistical hypotheses, except for a single metric on OS in the NICE group vs all others, which does not clearly address the stated aims.
- We thank you for raising this and apologise for the lack of clarity. We have now defined a null hypothesis for the primary aim (how compliant are patients with NICE guidelines) and secondary aims (that NICE group will have greater OS and reduced PFS than those without) in methods, sample size. We hope this adequately addresses the reviewers concerns.
- It is unclear, whether the authors have designed their study as a superiority or non-inferiority study. Based on the current statistical description, it seems the authors designed their study to document study failure to 5% significance assuming a survival HR of 1.35 in the two groups (NICE vs non-NICE). This is equivalent to a failed superiority assumption of adherence to the NICE guidelines. Rather the authors claim that these guidelines would arguably not result in significant changes in HR and thus the study should be conducted as an equivalence or non-inferiority study. The authors should clarify this and also elaborate on the statistical section and design aspect considerably.
- Thank you for highlighting this important issue. This study is indeed, a failed superiority assumption, relating to the NICE guideline adherence. This has been substantially modified in the protocol, to reflect that we are expecting to see superiority and a significant difference between the two groups.
- Regarding categorization of extent of resection, it is recommended that the authors adhere to RANO guidelines on post-op EOR.
- Thank you. We have modified the GTR definition in accordance with RANO criteria- in the UK, where the study is being conducted, scans are often employed between 48 and 72 hours after surgery due to resource constraints, therefore we have included post-operative scans up to 72 hours.